# Effect of Seminal Plasma Protein Fractions on Stallion Sperm Cryopreservation

**DOI:** 10.3390/ijms21176415

**Published:** 2020-09-03

**Authors:** Filipa Bubenickova, Pavla Postlerova, Ondrej Simonik, Jitka Sirohi, Jiri Sichtar

**Affiliations:** 1Department of Veterinary Sciences, Faculty of Agrobiology, Food and Natural Resources, Czech University of Life Sciences Prague, 165 00 Prague, Czech Republic; bubenickovaf@af.czu.cz (F.B.); postlerova@af.czu.cz (P.P.); j.sichtar@gmail.com (J.S.); 2Laboratory of Reproductive Biology, Institute of Biotechnology of the Czech Academy of Sciences, BIOCEV, 252 50 Vestec, Czech Republic; 3Department of Statistics, Faculty of Economics and Management, Czech University of Life Sciences Prague, 165 00 Prague, Czech Republic; sirohi@pef.czu.cz

**Keywords:** spermatozoa, biotechnology, cryobiology, phosphorylation, artificial insemination

## Abstract

Seminal plasma (SP) is the natural environment for spermatozoa and contains a number of components, especially proteins important for successful sperm maturation and fertilization. Nevertheless, in standard frozen stallion insemination doses production, SP is completely removed and is replaced by a semen extender. In the present study, we analyzed the effects of the selected seminal plasma protein groups that might play an important role in reducing the detrimental effects on spermatozoa during the cryopreservation process. SP proteins were separated according to their ability to bind to heparin into heparin-binding (Hep+) and heparin-non-binding (Hep−) fractions. The addition of three concentrations—125, 250, and 500 µg/mL—of each protein fraction was tested. After thawing, the following parameters were assessed: sperm motility (by CASA), plasma membrane integrity (PI staining), and acrosomal membrane integrity (PNA staining) using flow cytometry, and capacitation status (anti-phosphotyrosine antibody) using imaging-based flow cytometry. Our results showed that SP protein fractions had a significant effect on the kinematic parameters of spermatozoa and on a proportion of their subpopulations. The 125 µg/mL of Hep+ protein fraction resulted in increased linearity (LIN) and straightness (STR), moreover, with the highest values of sperm velocities (VAP, VSL), also this group contained the highest proportion of the fast sperm subpopulation. In contrast, the highest percentage of slow subpopulation was in the groups with 500 µg/mL of Hep+ fraction and 250 µg/mL of Hep− fraction. Interestingly, acrosomal membrane integrity was also highest in the groups with Hep+ fraction in concentrations of 125 µg/mL. Our results showed that the addition of protein fractions did not significantly affect the plasma membrane integrity and capacitation status of stallion spermatozoa. Moreover, our results confirmed that the effect of SP proteins on the sperm functionality is concentration-dependent, as has been reported for other species. Our study significantly contributes to the lack of studies dealing with possible use of specific stallion SP fractions in the complex puzzle of the improvement of cryopreservation protocols. It is clear that improvement in this field still needs more outputs from future studies, which should be focused on the effect of individual SP proteins on other sperm functional parameters with further implication on the success of artificial insemination in in vivo conditions.

## 1. Introduction

Successful cryopreservation of spermatozoa is still a challenging undertaking, especially that of stallions. It is estimated that only 20% of stallions produce semen that can be frozen well, and approximately 20–50% produce semen that is poorly freezable [1]. The production of frozen semen of an appropriate quality is of great importance in the horse-breeding industry, as it results in increased breeding and economic benefits, as well as the ability to preserve highly valuable breeds as genetic resources. There are different approaches for increasing the efficiency of the cryopreservation process [2,3,4]. 

The usual step used worldwide when increasing the post-thaw quality of stallion semen is to remove almost the entirety of the seminal plasma (SP) before cryopreservation [4]. However, the absence of SP in ejaculate processed for freezing raises questions, since different factors present in SP might affect the metabolism, motility, and freezability of spermatozoa [5]. Proteins present in SP directly interact with the plasma membrane (PM) of spermatozoa, preventing their premature capacitation and acrosome reaction [6]. If SP is removed before freezing, the availability of proteins important for PM functions is reduced. Thus, this step might lead to a disruption in PM function and decreased fertilization ability [7].

Sperm PM is exposed to two harmful events during cryopreservation, i.e., cold and osmotic shock [8]. These processes result in changes in lipid composition and consequently in decreased PM fluidity and permeability [9], leading to premature capacitation-like and acrosome reaction-like changes in sperm [10]. Currently, in bulls, boars, and rams, it seems that the mentioned negative effects of cryopreservation can be reduced by adding a certain amount of SP or particular components thereof to the suspension (spermatozoa + extender) during processing [6,11,12,13,14,15,16].

In stallions, only the effect of added SP on frozen-thawed sperm parameters has been studied, both before [17,18,19] and after freezing [20,21,22,23]. The results of these studies are very likely inconsistent due to the fact that the presence, absence, or different concentration of SP components, particularly proteins [24], differ between individual stallions [25], as well as between good and poor freezers [26].

The major proteins in stallion SP are horse seminal plasma proteins (HSP-1–8) [5,27]. They belong to three protein families: Fn-2 (fibronectin 2 type domain)-type proteins (HSP-1, HSP-2), cysteine-rich secretory proteins (CRISP; HSP-3), and spermadhesins (HSP-7) [27,28,29,30,31]. According to their ability to bind heparin, the SP proteins may be divided into heparin-binding (Hep+) and heparin-non-binding (Hep−) fractions. Proteins with heparin-binding capability include stallion proteins HSP-1, HSP-2, HSP-5, HSP-6, HSP-7, and HSP-8. Proteins HSP-3 (CRISP-3) and HSP-4 do not have the ability to bind heparin [27]. One possible way to overcome variability in SP effects is to use its isolated fractions or proteins [6]. With this strategy, it should be possible to take advantage of SP benefits and avoid the negative effects of some components [32], and to prevent variability in protein composition throughout SP [6].

In bulls, the effect of Hep+ proteins has been reported in frozen epididymal [13,14,15] and ejaculated spermatozoa [11,33]. It has been found that Hep+ proteins have a negative relationship with freezing ability, as well as with fertility [12,34] and with sperm quality parameters [15,35]. A negative association of Hep+ protein (HSP-1) with fertility was also found in horse seminal plasma [36,37]. On the other hand, the CRISP-3 protein, belonging to the Hep− fraction of stallion SP, is related to high semen freezability [32,37] and is a good marker of semen quality and fertility [36,38]. Moreover, it was suggested as a regulator of sperm elimination from the female reproductive tract [39]. In bulls and boars, the addition of Hep+ and Hep− proteins from SP had the opposite effect on viability, motility, and mitochondrial sperm activity [24], especially in boars, when the Hep− fraction had a positive effect in contrast to that of Hep+ proteins [6]. 

How do specific SP protein fractions affect the spermatozoa of stallions after thawing? In this study, we focus in detail on specific SP protein fractions, heparin-binding (Hep+), and heparin-non-binding (Hep−), especially regarding their assumed protective ability during the cryopreservation of stallion spermatozoa. 

## 2. Results 

The effect of seminal plasma protein fractions in three concentrations on stallion semen quality after thawing was evaluated on the basis of motility, sperm membrane integrity (viability), acrosomal integrity, and phosphorylation status.

### 2.1. Effect of SP Protein Fractions on Stallion Sperm Motility and Kinematic Parameters 

Results obtained from the CASA measurement are presented in Figure 1 and Figure 2. Total motility (TMOT) and progressive (PMOT) motility of spermatozoa in thawed samples after adding various concentrations of SP protein fractions did not significantly differ (*p* > 0.05). 

A more detailed analysis of individual kinematic parameters showed differences between the control and experimental groups (*p* < 0.05). Only the BCF parameter did not differ between tested groups (Figure 2). Parameter ALH was highest in the control group than all of the tested groups except the Hep−125 and Hep+125. Parameters LIN, STR, and WOB were lowest in the control group than all tested groups except Hep−500 and Hep+250. Parameters of velocity VAP and VSL were in the control group highest than groups Hep−250 and Hep+250. The value of parameter VCL was higher in the control group than groups Hep−250, Hep+ 125, and Hep+250.

The lowest ALH was in the Hep−250 group, which differed from the control and from all other concentrations except for Hep+500. The highest values of ALH, the control group, and Hep−125 differed significantly from all other concentrations except for Hep+125. In the parameter LIN, the dominant group was Hep+125, above all other samples. The LIN value in the Hep+250 group was significantly lower than those of the other samples and the control group. STR values were significantly higher in Hep+125 compared to all treatment groups except for Hep+500. STR values were significantly lower in the Hep+250 group than in every other group. VAP had the lowest values in the Hep−250 group compared to other treatment groups, except for Hep+250 and Hep+500. The highest value of parameter VAP was in the group Hep+125, significantly higher than groups Hep−250 and Hep+250. VCL values in the control, Hep−125, and Hep−500 groups were significantly higher than other treated groups except Hep+500. The lowest value of parameter VCL was in Hep−250 compared to other Hep− groups and Hep+250. VSL values were significantly lower in the Hep−250 and Hep+250 samples than in all other groups, except for the Hep+500 group. The highest VSL was in the group Hep+125 compared to groups Hep−250 and Hep+250. WOB was significantly highest in sample Hep+125 compared to all other groups. WOB in Hep−500 was significantly lower than in the Hep−125, Hep−250, and Hep+500 groups. WOB in the Hep+250 was lowest compared to that in the Hep−125, Hep−250, and Hep+500 groups. All differences between tested groups are shown in Appendix A.

### 2.2. Effect of SP Protein Fractions on the Distribution of Motile Spermatozoa to Subpopulations 

Additionally, the proportions of motile sperm subpopulations were analyzed (Figure 3, Appendix A). The highest proportions for the fast subpopulation were observed in the Hep+125 group and the control group, which were both statistically different from other groups (*p* < 0.05). The highest proportion of spermatozoa in the slow subpopulation and the lowest in the fast subpopulation was recorded in the Hep−250 group. The Hep−500 group had the highest proportion of medium to fast sperm. The highest proportion of the slow subpopulation in the Hep+ protein fraction samples was found in the 500 group, which also contained the lowest proportion of the fast subpopulation.

### 2.3. The Effect of SP Protein Fractions on Viability and Acrosomal Integrity

The presence of live cells (propidium iodide negative) and live cells with intact acrosomes (PNA-negative) was measured using flow cytometry. The effect of protein fractions in different concentrations on sperm plasma membrane and acrosomal integrity are presented in Table 1 and Figure 4. The Hep+125 group had a significantly (*p* < 0.05) higher concentration of cells with intact acrosomes compared to the Hep−125, Hep−500, and Hep+500 samples. Moreover, the Hep−250 sample had a higher concentration of spermatozoa with an intact acrosome, compared to the Hep−125 and Hep+500 groups (*p* < 0.05). Investigated sperm parameters in the experimental groups were not significantly different (*p* > 0.05) from the control group (Figure 4).

### 2.4. The Effect of SP Protein Fractions on Sperm Protein Phosphorylation

The protein tyrosine phosphorylation, indicating the sperm capacitation status, was measured using flow cytometry. In Figure 5, representative pictures of different sperm groups are shown: sperm without signal, low signal (equatorial segment, spots in the sperm head), medium signal (acrosomal or post-acrosomal region), and high signal (intense signal in the whole head with/without flagellum). The addition of SP protein fractions did not change the distribution of antibody signals in spermatozoa (Figure 6). When the protein phosphorylation was compared in spermatozoa after adding different concentrations of SP protein fractions, statistically significant (*p* < 0.05) differences were not observed. Nevertheless, there was a visible trend of increasing sperm without signal in samples after adding 250 µg/mL and 500 µg/mL of Hep+ SP fractions, indicating a low level of capacitated status.

## 3. Discussion

In the present study, we provide a more specific look at the effect of specific seminal plasma (SP) protein fractions on stallion cryopreserved spermatozoa. The main issue investigated in our present work was the protective capability of SP proteins against the detrimental effect of the freezing–thawing process. We separated SP proteins according to their ability to bind to heparin into heparin-binding (Hep+) and heparin-non-binding (Hep−) fractions. It has been reported previously that the influence of SP proteins on sperm is concentration-dependent [6,11,13]. In our study, we used concentrations (i.e., 0, 125, 250, and 500 µg/mL) based on a previous study that used a bovine model [40]. Results from different studies indicate that adding SP to proteins could have both positive and negative effects on the quality of cryopreserved spermatozoa [14,15,16,41]. The effect of SP proteins is very complex and has not been fully understood yet. Furthermore, seminal plasma is highly variable among species, within one species, between individuals, and among particular male ejaculates [6].

It is well known that cryopreservation decreases sperm motility and fertilization rate [42]. Although the effect of adding SP before or after the cryopreservation process on stallion sperm motility has been reported in several studies, results are inconsistent [17,18,20,21,22,23,43,44]. Different results may be caused by individual responses to SP addition and differences in its components [17]. To avoid individuality in SP components, we used separate protein fractions. Based on the ability to bind to heparin, we separated stallion SP protein into two fractions: heparin binding (Hep+) and heparin-non-binding (Hep−), which differed in protein composition [27]. In our study, adding both SP protein fractions to the freezing extender did not significantly impact on progressive (PMOT) or total motility (TMOT) after thawing. In a study on bulls, adding 20 µg/mL of Hep+ proteins to the freezing extender increased sperm motility pre-freezing as well as after freezing–thawing compared to the control group and other concentrations [33]. However, less protein was used than in our study. Kumar et al. [13] examined cauda epididymal spermatozoa and Hep− and Hep+ protein fraction in buffalo, which had a positive and negative effect on motility after thawing, respectively. Hep+ proteins had a positive effect on progressive motility of epididymal sperm only at the pre-freeze level [13], which was similar to the bovine model [15]. This effect could be explained by BSP (binder sperm proteins) in this fraction, which initiated cholesterol efflux and modulated the conformation of sperm membrane components, leading to a motility increase [15]. In boars, high concentrations of heparin-non-binding PSP-I protein are associated with reduced total and progressive sperm motility [45]. A recent study by Suárez et al., 2020 [46] proved that high concentrations of stallion SP decrease total and progressive motility, as well as some kinematic parameters of spermatozoa post-thaw. Our results showed neither detrimental nor positive effects of SP fractions on sperm motility; all treated groups gave results comparable with the control. However, adding protein fractions altered the kinematic parameters obtained using computer-assisted sperm analysis (CASA) measurement. The 125 µg/mL heparin-binding (Hep+) fraction provided the highest values (*p* < 0.05) regarding progressive movement, different velocities, and wobbles (LIN, STR, VAP, VSL, WOB). Higher values of kinematic parameters characterizing sperm speed and progressivity are related to higher quality and fertilization [47,48,49]. In donkeys, Taberner et al., 2009 [50] found a significant positive correlation between CASA parameters VAP, VCL, and ALH, and in vitro fertilization rates. In our previous study, adding stallion SP to sperm after thawing had a significant positive effect on kinematic parameters LIN, STR, VAP, VSL, and WOB (*p* < 0.01) [22]. Interestingly, that is consistent with the results of this study. However, different effects of total SP and a fractionated one might be caused by the processing of SP to isolate only the required proteins, when removal of ions and low-molecular-weight compounds may affect the sperm motility.

The analysis of distributing motile spermatozoa to subpopulations (slow, medium, and fast) could predict fertilization potential for frozen and thawed stallion sperm. Studying the distribution of motile sperm in subpopulations may be a suitable tool for improving routine analyses of stallion ejaculate, as it provides a more detailed view of its quality [47]. In individual kinematic parameters, we found that the highest number in the fast subpopulation and the lowest number of spermatozoa in the slow subpopulation were in the group treated with 125 µg/mL of Hep+ as well as in control group. On the other hand, Hep+ in the highest concentration (500 µg/mL) had the opposite effect on the percentage of sperm in subpopulations. Our previous study’s results showed that adding SP to the sperm sample post-thaw significantly increased the percentage of spermatozoa in the fast subpopulation [22]. According to a study by Ferraz et al., 2014 [51], the fast and progressive subpopulation correlated with in vitro fertilization ability. Thus, the subpopulation of sperm with maximum fertilization capacity had the highest velocities [47]. Moreover, Gibb et al., 2014 [52] emphasized the positive relationship between fast-moving sperm and pregnancy rates in mares and positive associations with higher cryoresistance [48]. 

SP proteins were reported in positive relation to membrane integrity as possible stabilizing compounds that act as protection for the sperm membrane [13]. It is suggested that SP contains decapacitation factors that bind it to the plasma membrane of sperm in order to maintain fertilization potential [6,53]. Most SP proteins maintain an uncapacitated sperm state, which prevents sperm agglutination and premature acrosome reaction, and further immune-suppresses the female reproductive tract [54]. In our study, plasma membrane integrity was not significantly affected after thawing by any protein fraction concentration. Interestingly, in the case of adding the Hep+ SP protein to a 500 µg/mL concentration, there was a negative trend on sperm viability after thawing. The expected protecting effect of some SP components against cryo-damage was not confirmed. It could be assumed that separation only on the based-on binding ability to heparin is not sufficient to separate positive and negative compounds and the mixture of proteins in these fractions might neutralize their effects [19]. In the bovine model, after cryopreservation, the effect of heparin binding (Hep+) to the protein fraction had a negative effect on plasma membrane integrity [33]. BSP, as part of the Hep+ SP protein fraction, showed a negative relationship with freezing ability and cattle fertility [12,34,35]. A negative association of HSP-1, a homolog to BSP-1, with fertility was also found in stallion SP [36,37]. BSPs may influence plasma membrane spermatozoa fluidity by increasing cholesterol efflux, ion fluxes [55,56,57], and phospholipid efflux, all of which contribute to membrane destabilization and increase sensitivity to cold shock and freezing [58].

An intact acrosome is a crucial part of a spermatozoon and is needed to achieve successful fertilization and other important sperm quality parameters after preservation. Positive correlations were obtained between acrosomal integrity and fertility value [59]. In our study, Hep+ protein fraction in concentrations of 125 µg/mL and 250 µg/mL resulted in the highest percentage of sperm with intact acrosomes (*p* < 0.05). On the other hand, insufficient protection of acrosome was recorded in samples of spermatozoa cryopreserved in the presence of high Hep+ SP fraction concentrations (500 µg/mL). This indicates that the effect of SP proteins on acrosomal integrity is concentration-dependent. We can deduce that that Hep+ proteins in higher concentrations in stallions may cause an acrosome reaction, which is presumed in bulls [33,40,60]. However, this is undesirable for sperm cryopreservation. Moreover, a previous study on buffalo cauda spermatozoa supports our results. When H^+^ proteins were added to the extender before freezing, there was a significant decrease in acrosomal integrity compared to the control [15]. On the other hand, acrosomal integrity was not affected by adding Hep+ or/and Hep− protein fractions in buffalo [13]. It was reported that Hep+ proteins in bulls (BSPs) stimulated cholesterol and phospholipid efflux from the sperm membrane, rendering the sperm plasma membrane sensitive to storage [41,56]. Spermatozoa are sensitive to oxidative stress because of their high polyunsaturated fatty acid content and relatively poor antioxidant defense [61].

Cryopreservation leads to capacitation-like changes, including protein phosphorylation, in sperm cells [10,62]. It was reported previously that capacitated sperm cells have a destabilized plasma membrane and are therefore sensitive to even small environmental stresses when compared to non-capacitated sperm plasma membranes [63]. Moreover, it has been suggested that longevity and low spermatozoa survival rate in the female reproductive tract are reduced by premature capacitation-related changes associated with cryopreservation, which have been demonstrated in bull, boar, and stallion sperm [23,64,65,66]. In cryocapacitated buffalo spermatozoa, similar tyrosine phosphorylated protein patterns as were capacitated in vitro [67]. Protein tyrosine phosphorylation in sperm has an important role in intracellular signaling, transport, and evaluation of phosphorylation, which can be used as a tool for monitoring sperm functionality after the capacitation process [55,68]. Based on the evaluation of phosphorylated proteins as one of the crucial known parameters for sperm capacitation, our study showed no significant effect of SP protein fraction on decreased cryocapacitation during the freezing–thawing of stallion sperm. Although there is no protocol which is able to fully capacitate spermatozoa [69] stallion spermatozoa also displayed an increase in protein tyrosine phosphorylation when capacitated in vitro [68]. Pommer et al., 2003 [70] reported an increase of phosphorylation in the mid and principal portions of sperm flagella when freshly ejaculated stallion sperm were capacitated. Vieira et al., 2013 [68] classified spermatozoa into two groups according to the localization of anti-phosphotyrosine staining (in the subequatorial region and flagellum), which were classified by fluorescence microscopy. In our study, four sperm groups were identified via image-based flow cytometric measurements: sperm without signal, low signal (equatorial segment, spots in the sperm head), medium signal (acrosomal or post-acrosomal region), and high signal (intense signal in the whole head with/without flagellum). The groups with no and low signal showed a low rate of cryocapacitation. After the addition of Hep+ protein fraction in the higher concentrations (250 and 500 µg/mL), anti-phosphotyrosine staining of spermatozoa demonstrated a positive trend in the groups with no and low signal. From these results, we can presume that in horse seminal plasma, the Hep+ proteins could belong to decapacitation factors. According to Centurion et al. (2003), SP contains decapacitation factors that prevent premature acrosomal reaction and increase the potential for fertilization. The decapacitating effect of SP is probably the result of proteins that are bound to the sperm surface, which prevents changes in the membrane that lead to capacitation and/or inhibit some signals of capacitation to other parts of the cell. Changes in frozen-thawed sperm shared many similarities with in vivo capacitation (Leahy et al., 2019). However, the cryopreservation does affect composition and organization of sperm plasma membranes that disturbs the lipid-lipid and lipid-protein interactions required for normal membrane function (Bailey et al., 2000). The fertilizing potential of sperm is reduced by their cryopreservation, and one reason could be the loss of surface proteins, as suggested by Lessard et al., 2000 [7].

Interestingly, this is contrary to in vitro studies in bovine and boar models, where it was shown that Hep+ proteins may serve as capacitation factors [40,60,71]. This discrepancy supports the recent knowledge about the huge difference in capacitation of stallion spermatozoa in vitro from other livestock species [40,71,72]. On the other hand, according to Suzuki et al., 2002 [73], boar whole seminal plasma addition (10% *v/v*) has a decapacitating effect. Interestingly, in boars, certain concentrations of heparin-binding proteins showed capacitation inhibitory activity during cooling and in vitro incubation. Nevertheless, during boar sperm cryopreservation, they were unable to prevent capacitation-like changes [74]. As reviewed by Leahy et al., 2019 [75], variability in seminal plasma composition and seminal plasma proteins across species complicates the task to reveal their importance and function on spermatozoa. Differences in accessory sex glands and mating strategies result in protein and composition of seminal plasma diversity [76].

Stallions differ from other mammalian species from the viewpoint of their capacitation. This may be related to varied composition in the sperm cell membranes and how the cholesterol regulates the fluidity and permeability of the lipid bilayers in the membrane. The variation in the capacitation rate of spermatozoa may be related to differences in the cholesterol/phospholipid ratio [77,78]. The individual predisposition of a stallion’s semen significantly influences the effectiveness of capacitation [79]. Reliable success has not yet been achieved in inducing the full capacitation state of stallion spermatozoa by specific capacitating triggers [69].

As mentioned above, our study, along with other studies, showed that the effect of seminal plasma protein fractions is concentration-dependent and is highly influenced by male individuality, especially in stallions. These results might be used for future experiments, where the stallion SP proteins will be identified, and the effect of individual proteins will be investigated. Our study provides data to approach the individual SP proteins in connection with their possible protection ability for spermatozoa during cryopreservation. It is clear that this field for improving the cryopreservation of stallion spermatozoa still needs more outputs from further future studies.

## 4. Materials and Methods

The study was carried out according to the Council Directive 98/58/EC, Act No. 154/2000 Coll., and Act of the Czech National Council No. 246/1992 Coll.

### 4.1. Processing of Seminal Plasma

Ejaculates were collected from stallions with proven good freezability of sperm in a certified equine reproduction center (ERC Ltd., Pardubice-Mnětice, Czech Republic). Ejaculates were obtained via a standard collection protocol using an open-type artificial vagina (CSU Model™ AV Case, Animal Reproduction Systems, Inc., Chino, CA, USA). Donors of seminal plasma (SP) were two stallions with long-term proven fertility and very good freezability (see more details in Sichtar et al., 2019 [22]. The obtained semen was centrifuged at 700 *g* for 10 min to remove spermatozoa and pooled. Afterwards, SP was centrifuged at 10,000 *g* for 10 min and stored at −80 °C until processing. 

### 4.2. Isolation of Seminal Plasma Protein Fractions

Seminal plasma proteins were separated on an affinity heparin–sepharose column (GE Healthcare, Uppsala, Sweden) into two fractions: heparin-binding (Hep+) and heparin-non-binding (Hep−), which were added after dialysis and lyophilization to the cryopreservation medium at the final three protein concentrations: 125 µg/mL, 250 µg/mL, and 500 µg/mL. The seminal plasma sample was thawed at room temperature and then centrifuged for final purification at 10,000 *g* for 5 min. The sample was then loaded onto a heparin–sepharose column, which was attached to a pump and an automatic collector. The pump flow rate for the Hep− fraction collection was 1.5 mL/10 min. When collecting the Hep− fraction, the column was aspirated with a PBS buffer (phosphate-buffered saline: 137 mM NaCl, 2.7 mM KCl, 10 mM Na_2_HPO_4_, 1.8 mM KH_2_HPO_4_, pH 7.4) to wash the column after aspirating the seminal plasma. After 150 min, the column was transferred to a 3 M NaCl solution in PBS, at a flow rate of 4 mL/10 min for 60 min, which released bound Hep+ proteins from the column. Separated samples in tubes were measured by a spectrophotometer (Biochrom, Libra S22, Fisher Scientific, Walthman, MA, USA) at 280 nm for protein content. Fractions with an absorbance above 0.03 were frozen and left in the freezer before protein fraction preparation. The individual protein fractions were subsequently dialyzed and lyophilized. Dialysis was done through membrane-cell dialysis tubing (MWCO 3500, SERVA, Heidelberg, Germany) for 48 h in a 0.2% acetic acid solution [80]. The acetic acid solution was exchanged at least three times over two days to remove all salts from the protein fraction solution. After dialysis, the fraction solutions were measured on a conductometer (Eutech ECTestr 11, OAKTON Instruments, Vernon Hills, IL, USA) to determine the salt content and dialysis efficiency. Before subsequent lyophilization, samples of protein fractions were placed in a freezer in lyophilization flasks. The frozen protein fractions were lyophilized (LYOVAC GT 2 E, FINN-AQUA) overnight until dry.

### 4.3. Sperm Collection and Freezing

Fresh ejaculate was collected from stallions at the national stud farm in Písek, s.p.o., Czech Republic (*n* = 7 stallions, two collections each) using an open-type artificial vagina (CSU Model™ AV Case, Animal Reproduction Systems, Inc., Chino, CA, USA). The ejaculates were subjectively evaluated, and only those which met the quality standards of the stud farm were used in this study. The collected ejaculate was immediately prediluted with skim-milk-based extender and centrifuged at 700 *g* for 15 min. Afterwards, the supernatant was removed and sperm pellets were divided into seven groups (2 mL/group): the control group was extended only with lactose–EDTA–egg yolk extender, which was privately manufactured by our laboratory (lactose, distilled water, glycerol, buffers, antibiotics, EDTA, and 20% (*v/v*) egg yolk). Other groups were extended with lactose–EDTA–egg yolk extender and treated with Hep+ or Hep− protein fractions at three concentrations: 125 µg/mL, 250 µg/mL, and 500 µg/mL. The final concentration of progressive spermatozoa per mL was 300 × 10^6^. Extended ejaculate was packed into 0.5 mL straws and horizontally frozen in a Styrofoam box 4 cm above the liquid nitrogen level for 15 min (Animal Reproduction systems, Chino, CA, USA), and then stored in liquid nitrogen.

### 4.4. Sperm Quality Parameters

#### 4.4.1. Motility Assessment

Each sample was thawed in a water bath (37 °C, 30 s), diluted on final concentration suitable for CASA measurement with Tyrode’s solution modified for sperm (Sp-TALP) (114 mM NaCl; 3.2 mM KCl; 25 mM NaHCO_3_; 0.34 mM NaH_2_PO_4_ × H_2_O; 10 mM sodium lactate; 2.0 mM CaCl_2_ × 2H_2_O; 0.5 mM MgCl_2_ × 6 H_2_O; 10 mM HEPES, redistilled Milli-Q water), and evaluated after 5 min of co-incubation at 37 °C in a water bath. 

Motility was measured with a CASA system (Nis-Elements, type 4.30, Laboratory Imaging, Prague, Czech Republic). The following parameters were evaluated: total motility (TMOT), progressive motility (PMOT), curvilinear velocity (VCL, µm/s), linear velocity (VSL, µm/s), average path velocity (VAP, µm/s), linearity (LIN, %), straightness (STR, %), wobble (WOB, %), amplitude of lateral sperm head motion (ALH, µm), and beat cross frequency (BCF, Hz). 

A 4 μL drop of each semen sample was placed in a 37 °C prewarmed Makler chamber (Sefi Medical Instrument, Haifa, Israel), and six fields per sample were evaluated at 100× magnification using a phase-contrast microscope (Eclipse E600; Nikon, Tokio, Japan) equipped with a heating plate prewarmed to 37 °C. The evaluation was based on the analysis of 41 consecutive digitized images, which were taken at a time lapse of 0.66 s with a camera at a frequency of 60 fps (DMK 23UM021; Imaging Source, Bremen, Germany). At least 200 trajectories were analyzed per field. The spermatozoa were considered motile when VAP >15 μm/s. The threshold values of STR and VAP for progressive motility were 70% and 15 μm/s, respectively. The distribution of spermatozoa into slow, medium, and fast subpopulations was based on the mean values of BCF, VAP, VCL, and VSL after a clustering analysis.

#### 4.4.2. Viability and Acrosomal Integrity

The ratio of live/dead sperm and acrosomal membrane status were determined using a flow cytometer (BD LSRFortessa^TM^ SORP, BD Biosciences, San Jose, CA, USA) using fluorescent dyes (i.r., propidium iodide (PI) dead-sperm staining, HOECHST 33342), which highlighted cell nuclei and differentiated the noncellular ejaculate components and fluorochrome-conjugated PNA lectin that bound to the acrosomal membrane. After thawing, samples were washed in Sp-TALP 5 min × 700 *g*, and a 5 µL sperm pellet was removed. Spermatozoa were diluted in 1 mL Sp-TALP and fluorescent dyes were added to the sample (HOECHST 12 µM, PI 12 µM, and PNA-FITC (1:500)). At least 30,000 cells per sample were evaluated (threshold events). For the FITC signal (PNA), a laser of 488 nm excitation and 525 nm emission filter was used. PI excitation had 561 nm and 610 nm; Hoechst excitation was 346 nm and 460 nm. The voltage was set for optimum signal acquisition according to the control samples with separate individual fluorescent dyes. Negative samples were prepared for the proper setting of the flow cytometer to eliminate possible autofluorescence prior to measurement.

#### 4.4.3. Sperm Phosphorylation Rate

The increased rate of phosphorylation of sperm proteins indicated an incipient capacitation and thus lessened spermatozoa viability. After thawing, sperm samples were washed twice (300 *g*, 5 min) in a warm phosphate-buffered saline (PBS) to remove the freezing extender. Sperm samples were then fixed in 3% formaldehyde for 10 min at room temperature. Samples were centrifuged two times in PBS (300 *g*, 5min) and permeabilized by 0.1% Triton-100 in PBS via incubation for 10 min at room temperature. After two PBS washings (300 *g*, 5 min) in primary antibody anti-phosphotyrosine, clone 4G10 (05-321, Millipore Sigma, Burlington, MA, USA) was added at a dilution rate of 1:300. A primary antibody was incubated in cell suspension overnight at 4 °C. For negative controls, the primary antibody was omitted. Subsequently, spermatozoa were washed and the PBS secondary antibody goat anti-mouse conjugated to tetramethylrhodamine isothiocyanate (GAR-Alexa568; Invitrogen, Carlsbad, CA, USA), which was used in a dilution of 1:400. Finally, suspension was incubated after adding Hoechst 33342 (Sigma-Aldrich) for 60 min in the dark at room temperature. Image-based flow cytometry (Amnis^®^ ImageStream^®^XMark II, AMNIS Luminex Corporation, Austin, TX, USA) was then performed and data were analyzed in the IDEAS software Version 6.0 (AMNIS Luminex Corporation) (Appendix A). The voltage was set for optimum signal acquisition according to control samples with separate individual fluorescent dyes. Negative samples were prepared for proper setting of the flow cytometer to eliminate possible autofluorescence prior to measurement.

### 4.5. Statistical Analysis

Data were statistically evaluated using STATISTICA software (version 12, StatSoft, Prague, CZ). The effect of the addition of seminal plasma fractions on TMOT, PMOT, selected kinematic parameters, viability, and data from flow cytometry measurements was evaluated via analysis of variance (ANOVA). Cluster analysis for sperm classification into subpopulations was used to evaluate motility. Euclidean distances were calculated from four variables: STR, VAP, VCL, and VSL. The chi square test was used to determine differences between subpopulations. Data were evaluated at *p* < 0.05 and described as the least-squares means (LSM) ± standard error of the means (SEM).

## Figures and Tables

**Figure 1 ijms-21-06415-f001:**
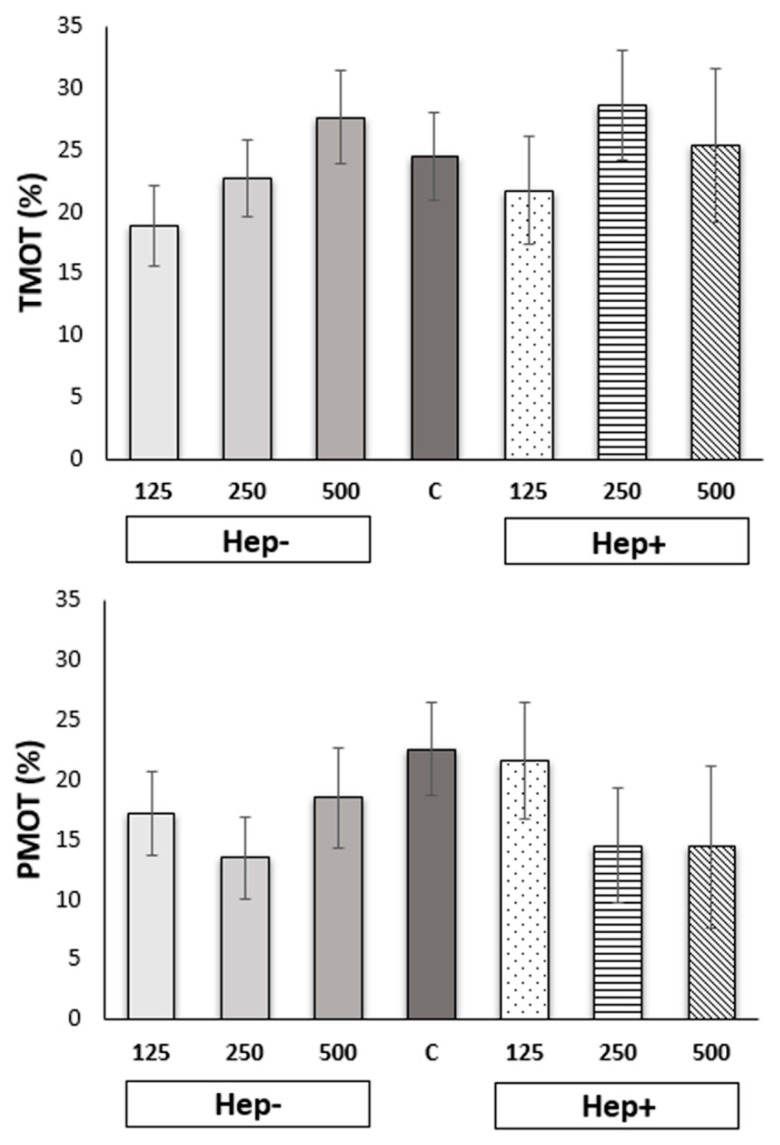
The effect of seminal plasma protein fractions in three concentrations on total motility (TMOT) and progressive motility (PMOT) after thawing. C: control group; Hep−: heparin-non-binding fraction in concentrations of 125 µg/mL, 250 µg/mL, and 500 µg/mL; Hep+: heparin-binding fraction in concentrations of 125 µg/mL, 250 µg/mL, and 500 µg/mL.

**Figure 2 ijms-21-06415-f002:**
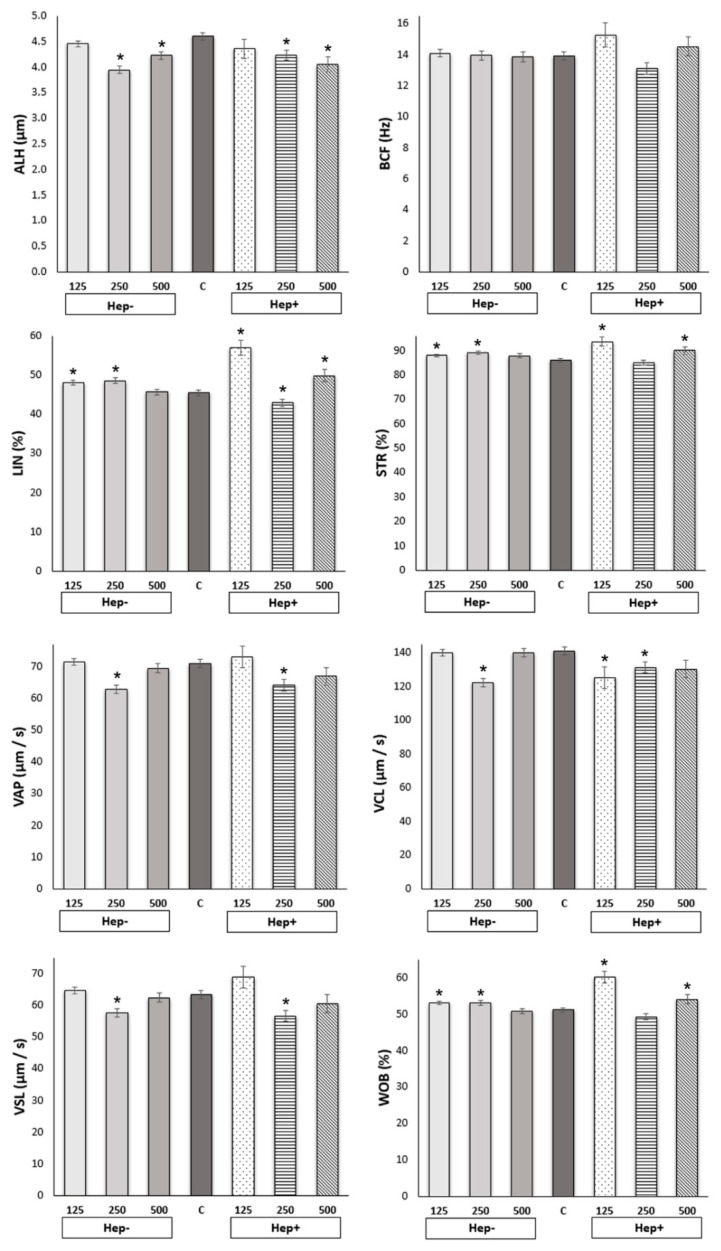
The effect of seminal plasma protein fractions in three concentrations on kinematic parameters (CASA). C: control group; Hep−: heparin-non-binding fraction in concentrations of 125 µg/mL, 250 µg/mL, and 500 µg/mL; Hep+: heparin-binding fraction in concentrations of 125 µg/mL, 250 µg/mL, and 500 µg/mL; CASA parameters: amplitude of lateral sperm head motion (ALH, µm), beat cross frequency (BCF, Hz), linearity (LIN, %), straightness (STR, %), average path velocity (VAP, µm/s), curvilinear velocity (VCL, µm/s), linear velocity (VSL, µm/s), wobble (WOB, %). Asterisk indicates statistical differences (** p* < 0.05) of experimental group vs. control.

**Figure 3 ijms-21-06415-f003:**
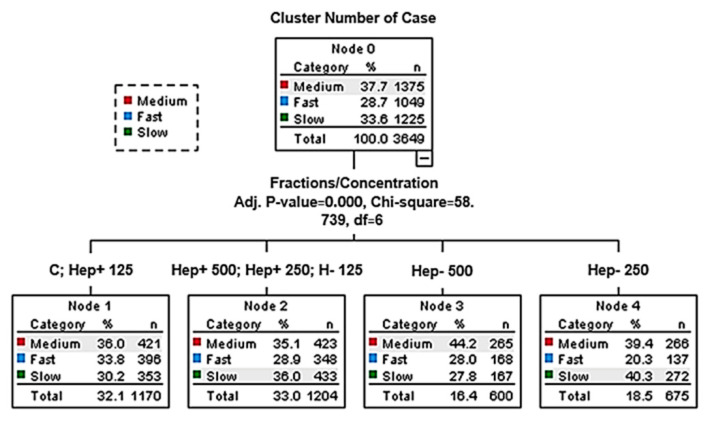
The effect of seminal plasma protein fractions in three concentrations on distribution of motile spermatozoa to subpopulations (slow, medium, fast). C: control group; Hep−: heparin-non-binding fraction in concentrations of 125 µg/mL, 250 µg/mL, and 500 µg/mL; Hep+: heparin-binding fraction in concentrations of 125, 250, and 500 µg/mL.

**Figure 4 ijms-21-06415-f004:**
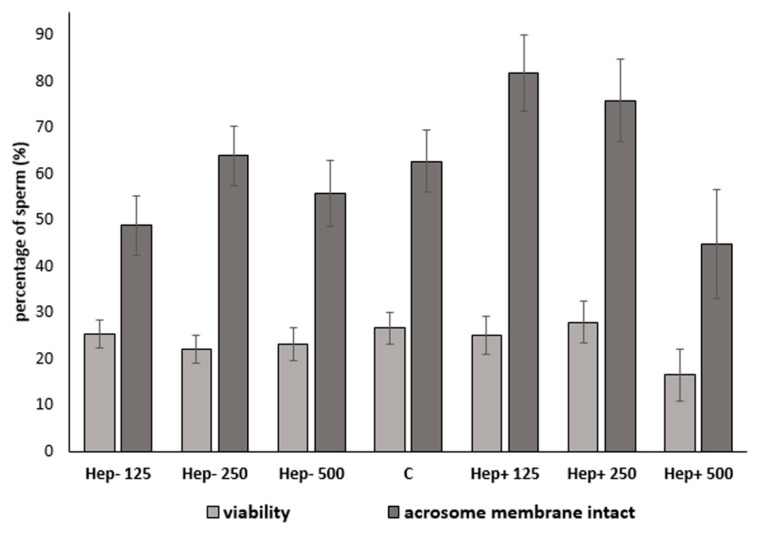
The effect of SP protein fractions on sperm viability and acrosomal integrity. C: control group; Hep−: heparin-non-binding fraction in concentrations of 125 µg/mL, 250 µg/mL, and 500 µg/mL; Hep+: heparin-binding fraction in concentrations of 125 µg/mL, 250 µg/mL, and 500 µg/mL.

**Figure 5 ijms-21-06415-f005:**
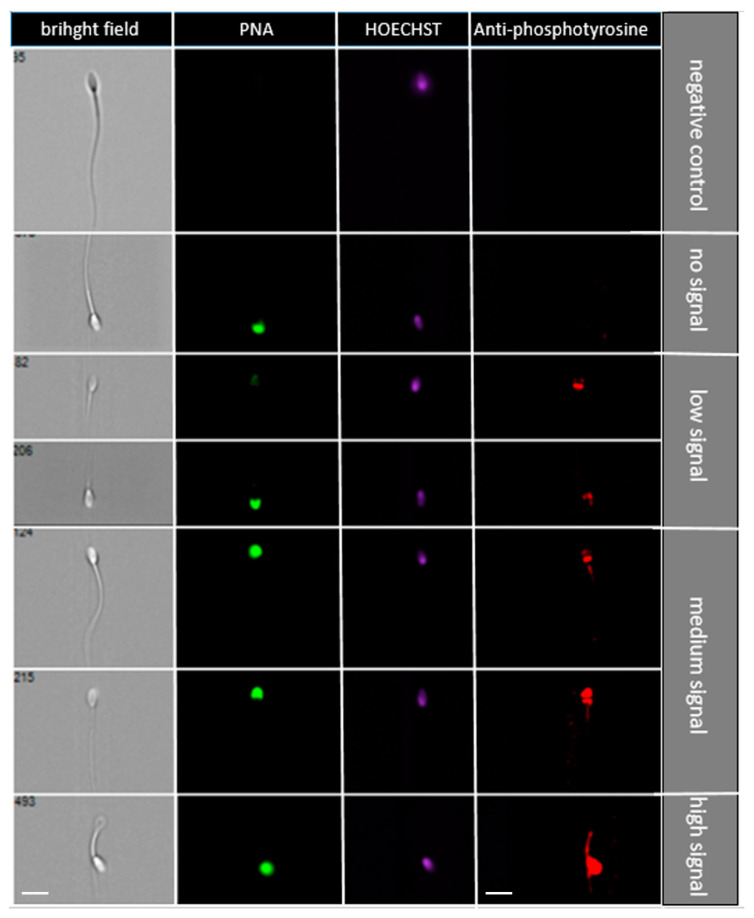
Representative images of different signaling from a flow cytometry measurement with an anti-phosphotyrosine antibody. Sperm groups: sperm without signal, low signal (equatorial segment, spots in the sperm head), medium signal (acrosomal or post-acrosomal region), and high signal (intense signal in the whole head with/without flagellum). Scale bar = 10 μm.

**Figure 6 ijms-21-06415-f006:**
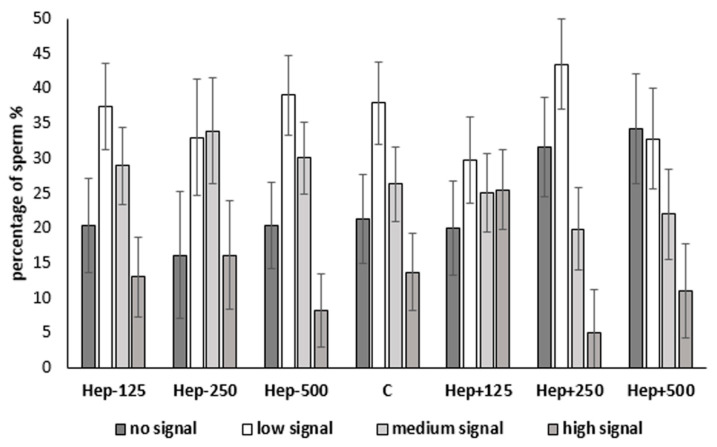
Effect of SP protein fractions in three concentrations on sperm phosphorylation signal intensity; C: control group; Hep−: heparin-non-binding fraction in concentrations of 125 µg/mL, 250 µg/mL, and 500 µg/mL; Hep+: heparin-binding fraction in concentrations of 125 µg/mL, 250 µg/mL, and 500 µg/mL.

**Table 1 ijms-21-06415-t001:** Effect of SP protein fractions on sperm viability and acrosomal integrity.

Protein Fraction	Concentration	PI: Live Cells	PNA: Live Cells with Intact Acrosomes
Hep−	125 µg/mL	25.36 ± 3.1	48.77 ± 6.4 ^2^
	250 µg/mL	22.03 ± 3.1	63.86 ± 6.4
	500 µg/mL	23.10 ± 3.5	55.78 ± 7.1 ^2^
C	0	26.58 ± 3.3	62.73 ± 6.7
Hep+	125 µg/mL	25.05 ± 4.0	81.87 ± 8.2 ^1,b^
	250 µg/mL	27.94 ± 4.4	75.88 ± 9.0 ^b^
	500 µg/mL	16.47 ± 5.7	44.77 ± 11.7 ^2,a^

^1,2^ and ^a,b^ indexes indicate significant differences (*p* < 0.05) from the highest (grey color) and lowest (blue color) values in all experimental and control groups.

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
