# Peer review of "Effect of Seminal Plasma Protein Fractions on Stallion Sperm Cryopreservation"

_ijms, 2020, doi:10.3390/ijms21176415_

Round 1
Reviewer 1 Report
Dear Author,
The aim of this study was to evaluate the effect, on stallion semen quality, of adding seminal plasma proteins fractions (H + and H -) to the extender before freezing.
This manuscript is reasonable well written. Results present new additional information and seem to be an original contribution, especially regarding to improving horse semen freezability. Conclusion are consistent with the results.
Below I have summarized some comments to address before this can be considered for publication:
Lines 91-94. Please, remove this paragraph. The aim of this study has been defined in the introduction section.
Lines 201-202. Please delete “after freezing-thawing”
Lines 207-208. Add “on” after impact.
Line 210. Please check the sentence “However, less proteins was used in our study”. In your study were used higher proteins concentrations than those in the mentioned study on bulls.
Lines 224-225. Please rewrite this sentence, there is a lack of the verb (i.e. are related).
Lines 234-235. In this study the highest number of spermatozoa in the fast subpopulation was found in the control group. That means that the addition of H + or H – SP proteins at different concentrations does not increase the percentage of fast spermatozoa in any group compared to control, so discussion about this aspect should be revised and rewritten.
Lines 244-263. Discussion in these paragraphs is too long and descriptive and it is focused on describing positive or negative effects of SP proteins on plasma membrane. It would be more interesting to discuss the reason why plasma membrane integrity was not affected by any of the added SP proteins concentrations.
Lines 247-275. In the mentioned study, addition of H + SP proteins to the extender before freezing improved acrosomal integrity so “decrease” should be change by “increase”.
Line 322. For a better understanding of this work, all materials and methods section should be placed after the introduction section.
Lines 323-324. Please, remove this paragraph. It has been repeated several times.
Line 328. Please, add what model of artificial vagina was used.
Lines 335-336. Seminal plasma was frozen/thawed and then centrifuged? or as it is previously said in lines 329-340, was it centrifuged and then frozen and stored at -80º C ?. Please, it should be clarified.
Line 355. What stallions were used to collect semen to freeze? with proven or unknown fertility? Similar o different from those used to process seminal plasma in order to isolate plasma protein fractions? It would be interesting to show in a table the average values for different seminal parameters after collection.
Lines 358-362. Please, add what was the final volume for each sample.
Lines 366-370. It would be useful to explain why after thawing samples were diluted in sp- TALP and at which concentration.
lines 402-403. It is not clear if all sperm quality parameters were immediately evaluated after diluting in sp-TALP. “after thawing, sperm samples were washed twice (300g, 5 min) in a warm PBS to remove freezing extender”.
Yours faithfully
Author Response
Lines 91-94. Please, remove this paragraph. The aim of this study has been defined in the introduction section.
Paragraph has been removed and rewritten.
Lines 201-202. Please delete “after freezing-thawing” - deleted
Lines 207-208. Add “on” after impact. - added
Line 210. Please check the sentence “However, less proteins was used in our study”. In your study were used higher proteins concentrations than those in the mentioned study on bulls.
Sentence has been rewritten. Thank you, it is very good point. That was mistake.
Lines 224-225. Please rewrite this sentence, there is a lack of the verb (i.e. are related). - rewritten
Lines 234-235. In this study the highest number of spermatozoa in the fast subpopulation was found in the control group. That means that the addition of H + or H – SP proteins at different concentrations does not increase the percentage of fast spermatozoa in any group compared to control, so discussion about this aspect should be revised and rewritten.
Thank you for your comment. We added the control group description.
Lines 244-263. Discussion in these paragraphs is too long and descriptive and it is focused on describing positive or negative effects of SP proteins on plasma membrane. It would be more interesting to discuss the reason why plasma membrane integrity was not affected by any of the added SP proteins concentrations.
We have shortened and rewritten this part of discussion. The possible protein interaction with plasma membrane is described also in other part of discussion (lns 310-319) because it was suggest by other reviewer.
Moreover, discussion related to individual concentration effects is very difficult in case of stallion. Studies which dealt with specific concentration of SP effect on spermatozoa were done on different model animals (bull, boar buffalo) and not always in relation with their cryopreservation.
Lines 247-275. In the mentioned study, addition of H + SP proteins to the extender before freezing improved acrosomal integrity so “decrease” should be change by “increase”.
Harshan et al. 2006. (Effect of buffalo seminal plasma heparin binding protein (HBP) on freezability and in vitro fertility of buffalo cauda spermatozoa) showed that addition of heparin binding proteins to freezing extender decreased acrosomal integrity after thawing. At pre-freeze level, the addition had not any effects. These results are showed clearly in Table 1. in the study. “A highly significant decrease in motility, viability and acrosomal integrity of spermatozoa was recorded in the HBP (heparin binding proteins) treated groups when compared to that in the control group”.
Line 322. For a better understanding of this work, all materials and methods section should be placed after the introduction section.
Section ordering is according to guidelines of IJMS.
Lines 323-324. Please, remove this paragraph. It has been repeated several times. - removed
Line 328. Please, add what model of artificial vagina was used. - added
Lines 335-336. Seminal plasma was frozen/thawed and then centrifuged? or as it is previously said in lines 329-340, was it centrifuged and then frozen and stored at -80º C ?. Please, it should be clarified.
We added this information to the text (lns 354-355, ln 361)
Line 355. What stallions were used to collect semen to freeze? with proven or unknown fertility? Similar o different from those used to process seminal plasma in order to isolate plasma protein fractions? It would be interesting to show in a table the average values for different seminal parameters after collection.
We used stallions with proven fertility from the national stud farm in Písek, s.p.o., Czech Republic, where housed stallions were represented different breeds selected for reproduction. The ejaculates were subjectively evaluated, and only these, which met quality standards of the stud farm have been used in this study. Donors of seminal plasma (ERC Ltd., Pardubice-Mnětice, Czech Republic) were stallions with proven fertility and very good freezability and the samples of seminal plasma were pooled.
We have added this information to the text (lns 352-353).
Lines 358-362. Please, add what was the final volume for each sample.
Each of 7 tested group (control and three H+ and three H- concentrations) from one collection from one stallion was in the final volume 2 ml to obtain 4 straws for further evaluation. We added this information to the text (ln 386).
Lines 366-370. It would be useful to explain why after thawing samples were diluted in sp- TALP and at which concentration.
Sp-TALP is one of the dilution medium used usually for handling with spermatozoa in in vitro conditions, especially for CASA, thus from our point of view it is not essential to write this detail. We moved the text to the motility assessment section (lns 396-399).
Lines 402-403. It is not clear if all sperm quality parameters were immediately evaluated after diluting in sp-TALP. “after thawing, sperm samples were washed twice (300g, 5 min) in a warm PBS to remove freezing extender”.
We agree, it was misleading. As the first part of sub-section “Motility assessment” has been modified (lns 397-401), now from our point of view details about dilution and washing are clear.
Reviewer 2 Report
The authors investigated the effects of seminal plasma protein fractions (heparin +/-) on the outcome of freezing/thawing of equine sperm. They could demonstrate a positive, dose-dependent effect on sperm function after freezing-thawing process. The study is well presented and its results are very interesting for equine breeding industry. It also contributes significantly to the recent research in this field and provides new insights for future research.
Author Response
The authors investigated the effects of seminal plasma protein fractions (heparin +/-) on the outcome of freezing/thawing of equine sperm. They could demonstrate a positive, dose-dependent effect on sperm function after freezing-thawing process. The study is well presented and its results are very interesting for equine breeding industry. It also contributes significantly to the recent research in this field and provides new insights for future research.
Thank you very much for your positive comment.
Reviewer 3 Report
The manuscript entitled "Effect of seminal plasma protein fractions on stallion sperm" by Bubenickova et al. examines the effects of seminal plasma on sperm motility and fertility. Recently it is known that seminal plasma proteins have sperm protection effects, and keep the sperm state as incapacitate. This manuscript shows that some proteins in the stallion seminal plasma had a significant effect on the kinematic parameters of cryopreserved spermatozoa. Although it is an immature study, the manuscript shows clear and interesting results. It is unsatisfactory that the fractionation of the SP proteins is only done on the basis of whether it binds to heparin or not. I would like to see more in-depth studies on individual proteins, considering the studies on other species (especially human and mouse). However, since it is expected that research on the stallion sperm have some difficulty, the studies in the manuscript may be acceptable. Thus, I consider that the manuscript is potentially worthy of publication in IJMS.
On the other hand, I still felt some experiments and texts should be improved, and these should be revised.
The authors need to clarify and reconsider the points listed below.
major points:
- The author stated that the aim of the study was to evaluate effect on stallion semen quality after thawing of adding seminal plasma (SP) protein fractions to the freezing extender. It is clear, and the studies are almost solid to evaluate it. On the other hand, as mentioned in the manuscript, the SP proteins (especially seminal vesicle secretions) is known to act as the decapacitation factor, that is, the incapacitation-preserving agent. I think that the cryoprotection and the incapacitation-preserving agent have a different mechanism of action, although some agent may have both effects. I hope that the author may study and discuss on the decapacitation effect.
- In mouse case, the SP proteins have an effect on sperm capacitation, and no effect on sperm motility of the incapacitated sperm. On the other hand, in the manuscript evaluated only sperm quality and motility only on thawed sperm. I do not know whether the method of sperm capacitation is established in stallion, but if could, the author should examine the effect of SP proteins on the sperm treated capacitated condition.
- Results 2.4 Sperm protein phosphorylation: It is interesting for me to examine distribution of Tyr-phosphorylated proteins in the sperm by immunostaining, but it is not standard experiment to evaluate the effect on sperm capacitation. If the authors intend to evaluate the effect of the SP proteins on sperm capacitation, the Tyr-phosphorylation should be examined by Western blotting. I recommend to add the experiment.
- There is no description about the approval of animal experiments in Materials and Methods section. It should be added.
Author Response
Reviewer 3
The manuscript entitled "Effect of seminal plasma protein fractions on stallion sperm" by Bubenickova et al. examines the effects of seminal plasma on sperm motility and fertility. Recently it is known that seminal plasma proteins have sperm protection effects, and keep the sperm state as incapacitate. This manuscript shows that some proteins in the stallion seminal plasma had a significant effect on the kinematic parameters of cryopreserved spermatozoa. Although it is an immature study, the manuscript shows clear and interesting results. It is unsatisfactory that the fractionation of the SP proteins is only done on the basis of whether it binds to heparin or not. I would like to see more in-depth studies on individual proteins, considering the studies on other species (especially human and mouse). However, since it is expected that research on the stallion sperm have some difficulty, the studies in the manuscript may be acceptable. Thus, I consider that the manuscript is potentially worthy of publication in IJMS.
On the other hand, I still felt some experiments and texts should be improved, and these should be revised.
The authors need to clarify and reconsider the points listed below.
major points:
- The author stated that the aim of the study was to evaluate effect on stallion semen quality after thawing of adding seminal plasma (SP) protein fractions to the freezing extender. It is clear, and the studies are almost solid to evaluate it. On the other hand, as mentioned in the manuscript, the SP proteins (especially seminal vesicle secretions) is known to act as the decapacitation factor, that is, the incapacitation-preserving agent. I think that the cryoprotection and the incapacitation-preserving agent have a different mechanism of action, although some agent may have both effects. I hope that the author may study and discuss on the decapacitation effect.
Great point, we have modified relevant part of discussion (lns 310-318)
- In mouse case, the SP proteins have an effect on sperm capacitation, and no effect on sperm motility of the incapacitated sperm. On the other hand, in the manuscript evaluated only sperm quality and motility only on thawed sperm. I do not know whether the method of sperm capacitation is established in stallion, but if could, the author should examine the effect of SP proteins on the sperm treated capacitated condition.
As described in detail in very recent review (Leemans et al. 2019), conventional in vitro fertilization (IVF) in horse is not reliably successful. In particular, stallion spermatozoa fail to penetrate the zona pellucida, most likely due to incomplete capacitation under in vitro conditions. In other mammalian species, specific capacitation triggers have been described; unfortunately, none of these is able to induce full capacitation in stallion spermatozoa.
Leemans, B.; Stout, T.A.E.; De Schauwer, C.; Heras, S.; Nelis, H.; Hoogewijs, M.; Van Soom, A.; Gadella, B.M. Update on mammalian sperm capacitation: how much does the horse differ from other species? Reproduction 2019, 157, R181-R197, doi:10.1530/rep-18-0541.
- Results 2.4 Sperm protein phosphorylation: It is interesting for me to examine distribution of Tyr-phosphorylated proteins in the sperm by immunostaining, but it is not standard experiment to evaluate the effect on sperm capacitation. If the authors intend to evaluate the effect of the SP proteins on sperm capacitation, the Tyr-phosphorylation should be examined by Western blotting. I recommend to add the experiment.
Thank you for a great idea and that we will surely include in the subsequent study. Here we were focused mainly on the monitoring of antibody localization and better quantifying of phosphotyrosine pattern of sperm cells by image-based flow cytometry. Additionally, in vitro capacitation in stallion sperm does not work completely reliably yet, so we could not compare the change in phosphorylated proteins caused by cryoconservation using WB.
During our experimental design preparation, we also made our decision based on studies listed below:
As stated in study (Pommer et al. 2003), after thawing, only immunofluorescence labelling was performed because frozen-thawed sperm samples had been cryopreserved in an egg-yolk-based extender. Therefore, these samples were unable to be electrophoresed and blotted adequately due to the residual yolk proteins from the extender.
In study Viera et al. (2013) was used same methodology of immunostaining to examine tyrosine phosphorylation on stallion spermatozoa. But in contrary to our study, evaluation tyrosine phosphorylation was performed only slides. In our study, we used image-based flow cytometry for better quantification moreover with possibility to localize signal of antibody.
Pommer, A.C.; Rutllant, J.; Meyers, S.A. Phosphorylation of protein tyrosine residues in fresh and cryopreserved stallion spermatozoa under capacitating conditions. Biology of Reproduction 2003, 68, 1208-1214, doi:10.1095/biolreprod.102.011106.
Vieira, L.A.; Gadea, J.; Garcia-Vazquez, F.A.; Aviles-Lopez, K.; Matas, C. Equine spermatozoa stored in the epididymis for up to 96 h at 4 degrees C can be successfully cryopreserved and maintain their fertilization capacity. Animal Reproduction Science 2013, 136, 280-288, doi:10.1016/j.anireprosci.2012.10.027.
- There is no description about the approval of animal experiments in Materials and Methods section. It should be added.
We added approvals of animal experiment (lns 346-347).
Reviewer 4 Report
In the second subsequent paper, the authors investigate the effect of post-thaw added seminal plasma in order to improve the quality of frozen and thawed sperm. Research of this kind is important to make the best use of the genetic material of donors with the most desirable traits in breeding. In recent years, a lot of studies have been published proving that sperm plasma is not only a passive medium for sperm transport, but also contains important active ingredients that affect the activity of gametes and their ability to fertilize the egg. Thus, the research aimed to identify such active components and the conditions under which their action is optimal is justified and important. However, the presented manuscript has some drawbacks which I present below and which should be corrected.
Major remarks
1. The authors do not sufficiently characterize the seminal plasma (SP) used for the research. The sperm samples of 7 stallions were analyzed. It is not known whether the SP originated from the same animals, from how many samples, whether they were pooled, or whether they were individual samples from the same animals. Detailed description is necessary here. The authors themselves admit that the composition of the SP presents high inter-subject variability, which may obviously affect the observed effect.
2. The results presented in the current work are not entirely consistent with those presented earlier [Animals 2019]. In the first study, which investigated the effect of unfractionated SP, the authors provide statistically significant increases in values for several analyzed parameters related to sperm motility: ALH, VAP, VCL, VSL. In the present study, the authors observe, at least in part of these parameters, a statistically significant decrease in the obtained values. It might seem that the SP preparations has lost their beneficial properties as a result of fractionation. In my opinion the authors should refer to these results in detail in the discussion.
3. There is no justification in the discussion as to why the authors decided to fractionate SP into heparin binding and non-binding fractions.
4. The Australian team of de Graaf and Leahy published some important works on the influence of SP on the condition of sperm in farmed animals after cryopreservation. Probably the authors should refer to the research of this team. It is worth mentioning in the context of the statement that the influence of SP on sperm properties can be both positive and negative. The properties of so called decapacitation factors and their species-dependent differences have been well reviewed by Tamara Leahy, also including discussion why in bulls the effect seems to be opposite to the other species.
Minor comments
* The H- and H + designation seems unfortunate and confusing to the reader (too much associated with the proton symbol). I would suggest using Hep +/-.
* Showing statistically significant differences on the figures is also not very clear. Since all differences are calculated in relation to the control group, there is no need to use two indeces (a, b), it is enough to mark statistically significantly different parameters with an asterisk.
* Isolation of SP. Apart from insufficient description of the material collection itself, the description of the further procedure is also not clear.
l. 334-5 Concentration of what? Lyophilized protein by weight or determined protein concentration? In the latter case method should be provided.
l. 337-8 usually flow rate and fraction volume is sufficient
l. 340 what is 10 mM 18 Na2HPO4?
l. 345 Fractions with absorbance above 0.03 mg / ml ??? Absorbance is dimensionless value, doesn’t provide protein concentration directly
l. 347 Why the protein was dialyzed to 4.5% acetic acid? What was the pH of the solution? These conditions may be harmful to the protein in 48h time period
* l. 200 cryopreservation may decrease fertilization rates, not fertility
Author Response
In the second subsequent paper, the authors investigate the effect of post-thaw added seminal plasma in order to improve the quality of frozen and thawed sperm. Research of this kind is important to make the best use of the genetic material of donors with the most desirable traits in breeding. In recent years, a lot of studies have been published proving that sperm plasma is not only a passive medium for sperm transport, but also contains important active ingredients that affect the activity of gametes and their ability to fertilize the egg. Thus, the research aimed to identify such active components and the conditions under which their action is optimal is justified and important. However, the presented manuscript has some drawbacks which I present below and which should be corrected.
Majorremarks
1. The authors do not sufficiently characterize the seminal plasma (SP) used for the research. The sperm samples of 7 stallions were analyzed. It is not known whether the SP originated from the same animals, from how many samples, whether they were pooled, or whether they were individual samples from the same animals. Detailed description is necessary here. The authors themselves admit that the composition of the SP presents high inter-subject variability, which may obviously affect the observed effect.
In this study, we used stallions with proven fertility from the national stud farm in Písek, s.p.o., Czech Republic, where housed stallions are represented different breeds selected for reproduction. These stallions were different from donors of seminal plasma housed in a certified equine reproduction center (ERC Ltd., Pardubice-Mnětice, Czech Republic). Donors of seminal plasma were two stallions with long-term proven fertility and very good freezability and the samples of seminal plasma were pooled.
We improved the description of the relevant part (lns 352 - 353) with reference to our previous study in the Animals journal (Sichtar et al., 2019).
The results presented in the current work are not entirely consistent with those presented earlier [Animals 2019]. In the first study, which investigated the effect of unfractionated SP, the authors provide statistically significant increases in values for several analyzed parameters related to sperm motility: ALH, VAP, VCL, VSL. In the present study, the authors observe, at least in part of these parameters, a statistically significant decrease in the obtained values. It might seem that the SP preparations has lost their beneficial properties as a result of fractionation. In my opinion the authors should refer to these results in detail in the discussion.
As reviewer wrote, in our previous study we used whole seminal plasma from our point of view it would not be appropriate to completely compare these results with current ones. Nevertheless, thank you for this important note, we modified relevant part of discussion (lns 232 - 234).
There is no justification in the discussion as to why the authors decided to fractionate SP into heparin binding and non-binding fractions.
Relevant information is in the first paragraph of the Discussion. Moreover, we partially modified last paragraph of this section regarding to the reviewer comment (lns 340-343).
- The Australian team of de Graaf and Leahy published some important works on the influence of SP on the condition of sperm in farmed animals after cryopreservation. Probably the authors should refer to the research of this team. It is worth mentioning in the context of the statement that the influence of SP on sperm properties can be both positive and negative. The properties of so called decapacitation factors and their species-dependent differences have been well reviewed by Tamara Leahy, also including discussion why in bulls the effect seems to be opposite to the other species.
We discussed the results of this important research group in the relevant part of discussion (lns 317-318; 326-329)
Minor comments
* The H- and H + designation seems unfortunate and confusing to the reader (too much associated with the proton symbol). I would suggest using Hep +/-.
All symbols have been rewritten throughout the manuscript.
* Showing statistically significant differences on the figures is also not very clear. Since all differences are calculated in relation to the control group, there is no need to use two indeces (a, b), it is enough to mark statistically significantly different parameters with an asterisk.
In figure 2 we marked significant differences to the control group with an asterisk.
* Isolation of SP. Apart from insufficient description of the material collection itself, the description of the further procedure is also not clear.
Relevant section of Material and methods has been modified
- 334-5 Concentration of what? Lyophilized protein by weight or determined protein concentration? In the latter case method should be provided.
We have specified it (ln 360)
l. 337-8 usually flow rate and fraction volume is sufficient
It was mistake we have left only the flow rate. (ln -368)
l. 340 what is 10 mM 18 Na2HPO4?
Thank you, it is mistake. We have removed number 18.
- 345 Fractions with absorbance above 0.03 mg / ml ??? Absorbance is dimensionless value, doesn’t provide protein concentration directly
Thank you, it is mistake. We have removed mg / mL. (ln 371)
- 347 Why the protein was dialyzed to 4.5% acetic acid? What was the pH of the solution? These conditions may be harmful to the protein in 48h time period
Thank you for this comment. It is mistake. We used 0.2% acetic acid according to Manaskova et al., 1999, where boar SP proteins were fractionated and used for binding studies. The pH of this solution is acidic, because SP proteins have a tendency to precipitate out from solution at higher pH. We revised this and added reference (lns 373-374)
* l. 200 cryopreservation may decrease fertilization rates, not fertility - rewritten
Round 2
Reviewer 3 Report
The revised manuscript incorporated the reviewers’ indications, thus it was refined and improved. The data shown in the manuscript are sound, and it may be suitable for IJMS.
Reviewer 4 Report
The authors have addressed in detail all the issues included in the first review. Therefore the improved manuscript in the current form may be accepted for publication